# Proximity-driven site-specific cyclization of phage-displayed peptides

Libby Brown[1,2], Aldrin V. Vidal [1], Ana Laura Dias [3], Tiago Rodrigues [3], Anna Sigurdardottir [2], Toby Journeaux[1], Siobhan O'Brien[2], Thomas V. Murray [2], Peter Ravn[2,5], Monika Papworth[2] & Gonçalo J. L. Bernardes [1,4] ✉

Cyclization provides a general strategy for improving the proteolytic stability, cell membrane permeability and target binding affinity of peptides. Insertion of a stable, non-reducible linker into a disulphide bond is a commonly used approach for cyclizing phage-displayed peptides. However, among the vast collection of cysteine reactive linkers available, few provide the selectivity required to target specific cysteine residues within the peptide in the phage display system, whilst sparing those on the phage capsid. Here, we report the development of a cyclopropenone-based proximity-driven chemical linker that can efficiently cyclize synthetic peptides and peptides fused to a phage-coat protein, and cyclize phage-displayed peptides in a site-specific manner, with no disruption to phage infectivity. Our cyclization strategy enables the construction of stable, highly diverse phage display libraries. These libraries can be used for the selection of high-affinity cyclic peptide binders, as exemplified through model selections on streptavidin and the therapeutic target αvβ3.

Small molecules and large biologics, such as antibodies, have traditionally been the most pursued therapeutic formats in drug development. However, cyclic peptides provide a promising alternative, due to their high proteolytic stability, high target binding affinity/specificity, and low toxicity[1,2]. Phage display technology enables the construction of cyclic peptide libraries containing up to $10^{12}$ different phage variants and provides a powerful tool for identifying de novo cyclic peptide ligands that bind with high affinity to an array of therapeutic targets (e.g., cyclic peptides that inhibit protein-protein interactions for drug discovery applications)[3]. Whilst several different methods have been established to cyclize phage-displayed peptides, from disulfide bond formation to the use of chemical crosslinkers and non-canonical amino acids, each has its advantages and disadvantages (Fig. 1a).

To promote disulfide bond formation, a randomised peptide sequence is flanked on either side by two cysteine residues. Due to the oxidising environment within the bacterial periplasm, the peptide spontaneously cyclizes during filamentous phage assembly. No further chemical treatment is needed if the phage-displayed peptides are retained in non-reducing conditions. Disulfide-cyclized peptides often have binding affinities in the millimolar to picomolar range (typically within the micromolar range), and often have higher selectivity than their linear counterparts, due to their higher conformational stability both on phage and when synthesised. However, disulfide-cyclized peptides have a limited application in vivo, as they cannot withstand reducing cellular environments; the disulfide bonds are susceptible to reduction and disulfide exchange reactions[4,5].

An alternative approach is to use small molecule linkers to covalently connect two cysteine residues to yield cyclic peptides with significantly improved in vivo stability. For example, the use of a 1,3,5-tris(bromomethyl)benzene (TBMB) linker in phage display facilitated

[1]Yusuf Hamied Department of Chemistry, University of Cambridge, Cambridge, UK. [2]Biologics Engineering, Oncology R&D, AstraZeneca, The Discovery Centre; Cambridge Biomedical Campus, Cambridge, UK. [3]Instituto de Investigação do Medicamento (iMed), Faculdade de Farmácia, Universidade de Lisboa, Lisboa, Portugal. [4]Instituto de Medicina Molecular João Lobo Antunes, Faculdade de Medicina da Universidade de Lisboa, Lisboa, Portugal. [5]Present address: Department of Biotherapeutic Discovery, H. Lundbeck A/S, Valby, Denmark. ✉e-mail: gb453@cam.ac.uk

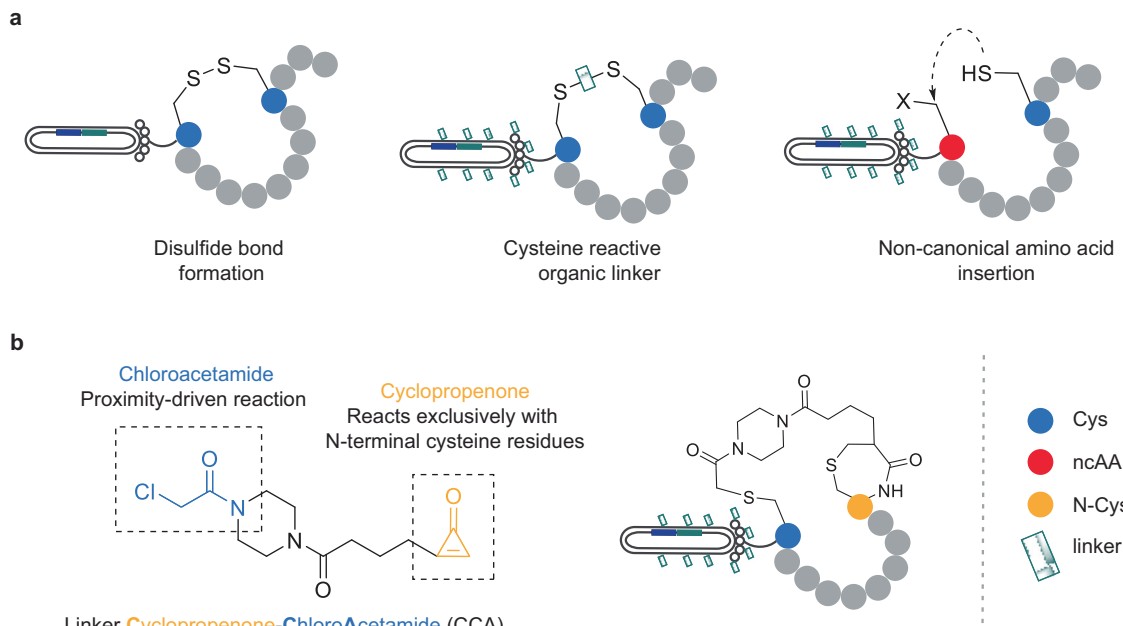

**Fig. 1 | Existing and proposed strategies for the cyclization of phage-displayed peptides. a** Several different methods have been established to cyclize phage-displayed peptides, from disulfide bond formation to the use of chemical crosslinkers and non-canonical amino acids, but each has its disadvantages. **b** Our proposed cyclopropenone-based proximity-driven approach for the site-specific cyclization of phage-displayed peptides.

the selection of a potent inhibitor of human plasma kallikrein with an IC$_{50}$ value of 1.7 nM. For reference, the IC$_{50}$ of the selected linear peptide without TBMB modification was greater than 10 μM, which highlights the superior therapeutic potential of cyclic peptides compared to their linear counterparts[6]. In more recent years, alternative chemical linkers have been discovered containing either hydrogen bond acceptors or donors. The formation of hydrogen bonds between the peptide and the chemical linker produces highly compact and rigid phage-displayed peptides with superior binding selectivity[7].

Cysteine reactive chemical cyclization linkers suffer from several intrinsic limitations when used in filamentous phage display systems with phage coat protein pIII[8,9]. Firstly, cysteine residues found within pIII (eight in total) can be modified, which affects the viability and or/ infectivity of the phage particles. To overcome this problem, an engineered class of bacteriophage can be used with all cysteine residues removed from pIII. However, these disulfide-free phages suffer from ~100-fold reduced infectivity compared to the wild-type phage and thus, library size/ diversity is limited[9]. Secondly, the concentration of modifying linker must be controlled to prevent cross-linking of phage coat proteins via non-specific lysine modification[6]. This can result in slow reaction times and/or incomplete cyclization. Finally, these linkers must be symmetrical to prevent the formation of unwanted regioisomers. This prevents the incorporation of asymmetric molecular scaffolds, which are often found in natural macrocyclic peptides[10].

As an alternative to using cysteine pairs and chemical crosslinkers, genetically encoded libraries of cyclic peptides can be engineered to contain non-canonical amino acids (ncAAs). For example, if a randomised peptide is flanked by a cysteine residue and a cysteine reactive non-canonical amino acid (ncAA), introduced on production via amber-suppression methodology, cyclization occurs through the spontaneous formation of a non-reducible bridge between the cysteine and ncAA residue. Although this approach overcomes the problems associated with traditional cysteine reactive cross-linkers, ncAA insertion often involves complex production and expression levels tend to be low, which in turn impacts library size and diversity[9–11].

In this work, we report the development of a heterobifunctional **c**yclopropenone-**c**hloro**a**cetamide (CCA) proximity-driven cyclization linker (Fig. 1b). Linker CCA can efficiently cyclize phage-displayed peptides in a site-specific manner without disruption to phage infectivity and/or viability. This specificity is enabled by the exquisite selectivity and fast kinetics ($k = 67$ M$^{-1}$. s$^{-1}$) of the cyclopropenone towards 1,2-amino thiols, which places the chloroacetamide within proximity for an intramolecular reaction with the internal cysteine[12]. We were able to validate our approach through (A) the construction of a stable and highly diverse CCA-cyclized CX$_{10}$C peptide phage display library, (B) the selection of a novel class of cyclic peptide binders to streptavidin, and (C) the identification of high-affinity (low nM) binders to the therapeutically relevant target, integrin αvβ3[13].

## Results

### The design and synthesis of linker cyclopropenone-chloroacetamide (CCA)

We proposed that bioorthogonal N-terminal cysteine chemistry developed for the site-specific modification of proteins and peptides could be used to tackle the conjugation-site selectivity problems associated with traditional cyclization linkers. Although several chemical approaches have been implemented to generate peptide phage display libraries cyclized through an N-terminal to internal cysteine linker, each has its disadvantages: native chemical ligation leads to a significant reduction in phage viability, likely due to non-specific linker conjugation[14]; 2-cyanobenzothiazole (CBT) chemistry suffers from selectivity issues as the CBT functional group can react with simple thiols (albeit reversibly)[15,16]; and 2-((alkylthio)(aryl)methylene)malononitrile (TAMM) phage cyclization requires a large excess of the linker, which is time-consuming and difficult to synthesise[17].

Alternative bioconjugation reagents are required for fast and selective N-terminal to internal cysteine cyclization, without disruption to phage infectivity and/or viability. To this end, a bifunctional linker was designed to contain cyclopropenone (CPO) at one end of the molecule, and chloroacetamide at the other (Fig. 1b). Cyclopropenone reagents have been shown to label N-terminal cysteine residues on proteins and peptides in a site-specific manner. The reaction

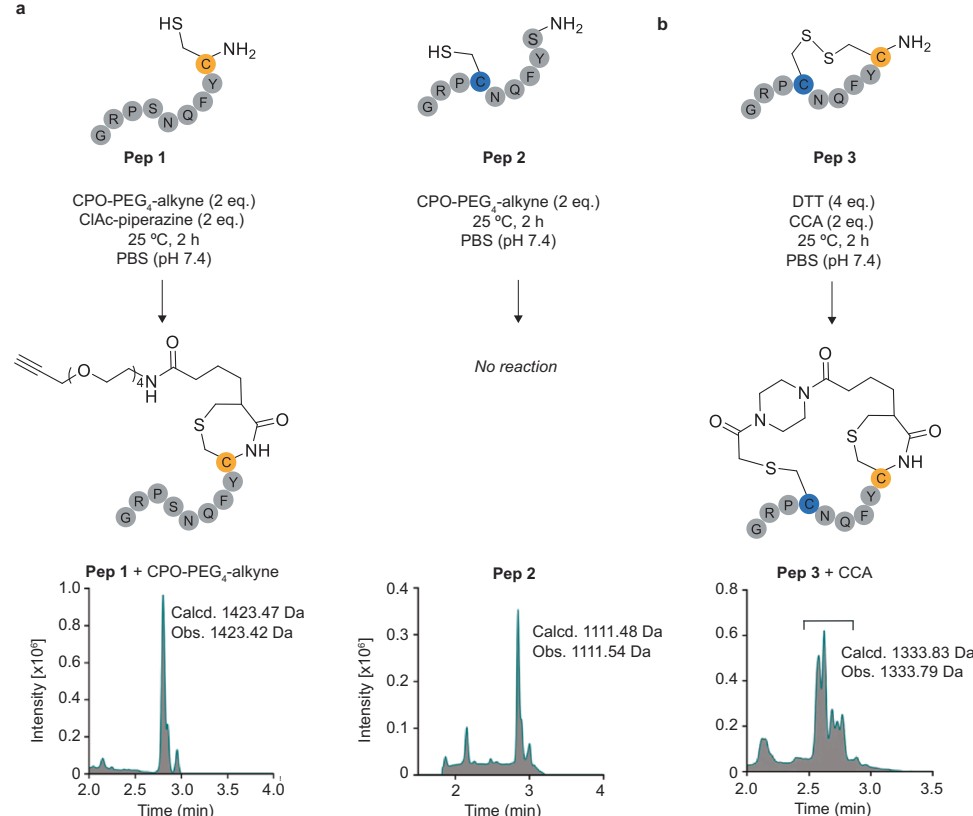

**Fig. 2 | CPO reagents selectively modify N-terminal cysteine residues.** Chloroacetamide reagents only undergo intramolecular reactions when positioned in close proximity to a free cysteine residue. Chromatograms show absorbance at 280 nm. **a** Treatment of peptide 1 (**Pep 1**) with 1:1 mix of CPO-PEG$_4$-alkyne: piperazine-ClAc yields a single CPO-modified product after 2 h at 25 °C. No reaction is observed between CPO-PEG$_4$-alkyne and peptide 2 (**Pep 2**). **b** Full conversion to CCA-cyclized peptide 3 (**Pep 3**) is observed after 2 h at 25 °C in the presence of DTT. The four chromatogram peaks correspond to the four different stereoisomers.

proceeds with high efficiency and kinetics comparable to those of maleimide reactions in mild conditions (aqueous buffer, pH 7, 4–25 °C) and is not affected by the presence of solvent-exposed cysteine residues, either on the same sequence or in a mixture of proteins (mechanism depicted in Supplementary Fig. 1). CBT-based chemistry is arguably the state-of-the-art method for N-terminal cysteine modification. However, recent reports have demonstrated that CPO reagents exhibit similar reaction kinetics to their CBT analogues, and have significantly improved site-specificity for N-terminal cysteine residues[12].

Chloroacetamides (ClAcs) react very slowly with free thiols at neutral pH. However, the reaction rate can be significantly improved by positioning the ClAc close to a free cysteine residue for a proximity-driven intramolecular reaction[18]. To demonstrate the specificity of CPO-based reagents for N-terminal cysteines, we designed two peptides based on vasopressin (pep3). We replaced either the cysteine at position 6 or the N-terminal cysteine with a serine residue to generate peptides 1 (**C**YFQNSPRG) and 2 (SYFQN**C**PRG) respectively. Treatment of peptide 1 with a 1:1 mixture of CPO-PEG$_4$-alkyne and piperazine-ClAc yielded 100% conversion to the CPO, rather than ClAc modified product (Fig. 2a). Treatment of peptide 2 with CPO-PEG$_4$-alkyne under the same conditions led to no appreciable modification (Fig. 2b), whereas vasopressin was readily modified (single addition) (Supplementary Fig. 2), highlighting the selectivity of CPO for N-terminal rather than internal cysteine residues.

Having confirmed the selectivity of CPO reagents for N-terminal cysteine residues and the low reactivity of chloroacetamide reagents, we progressed to the synthesis of linker CCA. A rigid piperazine ring was chosen to connect the CPO and ClAc functionalities to further reduce conformational flexibility. Commercially available 5-hexynoic acid was first converted to the corresponding pentafluorophenol (PFP) ester **1**. Cyclopropenation of **1** yielded the CPO-modified activated ester CPO-PFP (**2**). Amine **3** (ClAc-piperazine) was prepared via reaction of 1-Boc-piperazine with chloroacetyl chloride, followed by Boc deprotection. Upon coupling of the two intermediates amine **3** and CPO-PFP, linker CCA was obtained in high yield. In a similar manner, CPO-PFP was coupled with commercially available amine-PEG-biotin to yield CPO-PEG$_2$-biotin (Supplementary Fig. 3).

### Proof-of-concept peptide and protein cyclization using CCA

To test its reactivity and selectivity, CCA was conjugated to a model peptide (P3-vasopressin, **C**YFQN**C**PRG), containing both an N-terminal and internal cysteine residue. First, vasopressin was reduced with DTT (4 equiv.) for 1 h at 25 °C to expose two free cysteine residues at positions 1 and 6. DTT was used in preference to TCEP as it has been shown not to react with CPO reagents, even when added in huge excess[12]. Next, CCA (2 equiv.) was added and > 95% conversion to the CPO-cyclic peptide was observed after 2 h at 25 °C (Fig. 2b).

The results obtained for peptides **1**–**3** were promising but a more complex model system was required to confirm the regio-/chemoselectivity of linker CCA in the context of a full phage particle. Protein III (pIII), displayed on the surface of the phage particle (5 copies per phage), contains three disulfide bonds within the D1/D2 domains (C7–C36, C46–C53 and C188–C201)[8]. We hypothesised that linker CCA could selectively conjugate to the peptide cysteine residues whilst sparing those found within pIII, because of the selectivity and fast kinetics of the reaction of CPO with the N-terminal cysteine. This would allow for the use of wild-type rather than disulfide-free pIII, and phage infectivity would remain high.

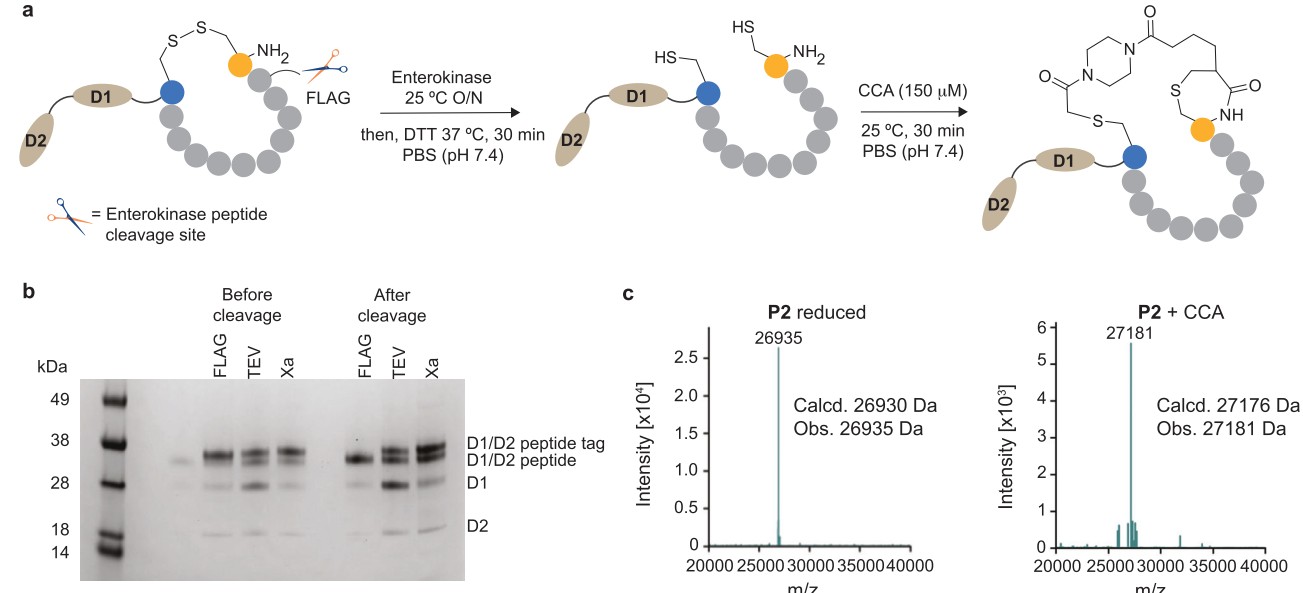

**Fig. 3 | CCA-mediated cyclization of peptides fused to the D1/D2 soluble domains of phage-coat protein pIII. a** A schematic representation of the treatment of P1-FLAG D1/D2 fusion protein with enterokinase cleavage enzyme, reduction with DTT and cyclization with linker CCA. **b** SDS-PAGE analysis of the cleavage of a FLAG, TEV or Factor-Xa peptide tag appended to the N-terminus of the displayed peptide. Representative of one of three independent replicates. **c** LC-MS analysis of P2 D1/D2 fusion protein before and after treatment with 150 μM CCA for 2 h at 25 °C.

To test this hypothesis, three control constructs (P1-FLAG, P1-TEV and P1-Xa) were designed, incorporating different peptides at the N-terminus of the soluble D1/D2 domains of pIII. An amber stop codon was inserted at the C-terminus of D2 to allow for the expression of the soluble D1/D2-peptide fusion and a His-tag for purification. Peptide P1 (CTTEEPYLVCWL) followed by a short spacer (AAA) was fused between the pelB leader sequence and the N-terminus of pIII D1. Peptide P1 is derived from a matrix metallopeptidase 12 (MMP12) binding peptide, identified from phage display selections, which expresses well as a D1/D2 fusion protein. The PelB leader sequence drives the secretion of pIII into the periplasm for phage packaging. Once in the periplasm, the PelB leader is cleaved by endogenous *E. coli* leader peptidase, which can be inefficient if a cysteine residue is placed immediately adjacent to the leader sequence[19]. To allow for the efficient production of proteins with N-terminal cysteine, short cleavable peptides (cysteine-free) were added between the leader and the N-terminus of the P1 sequence. These included a FLAG tag (P1-FLAG, cleavable by enterokinase), a TEV recognition site (P1-TEV) and a Factor Xa recognition site (P1-Xa). Following expression of the D1/D2-fusion protein, enterokinase, TEV, or Factor Xa cleavage removes all upstream residues of the CTTEE-PYLVCWL, to expose the second internal cysteine as the new N-terminus.

Fusion proteins P1-FLAG, P1-TEV and P1-Xa were expressed and purified using Ni²⁺-affinity chromatography. The successful expression, secretion and purification of all three fusion proteins were confirmed by SDS-PAGE and LC-MS analysis. Proteins P1-FLAG, P1-TEV and P1-Xa were treated with their respective proteolytic enzymes and cleavage efficiency was monitored by SDS-PAGE and LC-MS analysis. Of the three approaches, cleavage of the FLAG tag with enterokinase enzyme was the most efficient, with 100% cleavage observed after 16 h at 25 °C. In comparison, only 20% conversion of protein P1-Xa to the N-terminal cysteine exposed D1/D2-peptide fusion was observed under the same conditions. The TEV cleavage site proved to be the least efficient, with no cleavage observed after 16 h at 25 °C (Fig. 3b).

Having established the Flag-tag system, we proceeded to express as D1/D2 fusions a series of peptides, P2 to P6, all containing an N-terminal cysteine (Table 1). Fusion protein P2-FLAG was expressed, purified with anti-FLAG magnetic resin, and cleaved with enterokinase to yield protein P2. To optimise reaction conditions for conjugation, P2 was reduced with DTT and incubated with a 5–50-fold excess of linker CCA. Full conversion to the cyclized product was observed with >10-fold excess CCA after 2 h at 25 °C (Fig. 3c). In contrast, performing the same procedure under optimised conditions with FLAG-capped P2 (no enterokinase cleavage), resulted in no appreciable modification, again highlighting the site-specificity of linker CCA for free N-terminal cysteines.

To test the effect of peptide length on the efficiency of cyclization, proteins P3, P4 and P5 were expressed with an N-terminal Flag tag, purified on anti-FLAG resin, and cleaved with enterokinase enzyme (Fig. 3d). All proteins were treated with 150 μM CCA and modification was monitored via LC-MS. Complete conversion to the CCA-cyclized product was observed for all three constructs. Furthermore, all CCA-cyclized constructs were found to be stable in a redox buffer containing 2 mM glutathione (Supplementary Fig. 11).

As an additional control, protein P6 with no FLAG tag appended to the N-terminus was expressed and purified via Ni-²⁺ affinity chromatography. As expected, a high degree of product heterogeneity, mainly caused by peptide clipping, was observed, and a ~ 3:1 ratio of D1/D2: D1/D2-pep was obtained. In the absence of a Flag-tag at the N-terminus of this tag-free control, the intact material could not be purified. Nevertheless, the above mix of D1/D2 and D1/D2-pep was treated with 150 μM CCA for 2 h at 25 °C to monitor CCA specificity. As shown in Supplementary Fig. 10, protein P6 is fully converted to its cyclic

**Table 1 | Displayed amino acid (AA) sequences of pep-D1/D2 fusion proteins P1-P6**

| Pep-D1/D2 fusion | AA sequence of displayed peptide |
|---|---|
| P1 | CTTRRPYLVCWL |
| P2 | CGGSGGC |
| P3 | CGSGGSGC |
| P4 | CQPHPGQTC |
| P5 | CPEGYILDDGFCTDIDE |
| P6 | CYKLAEGDKYYIC |

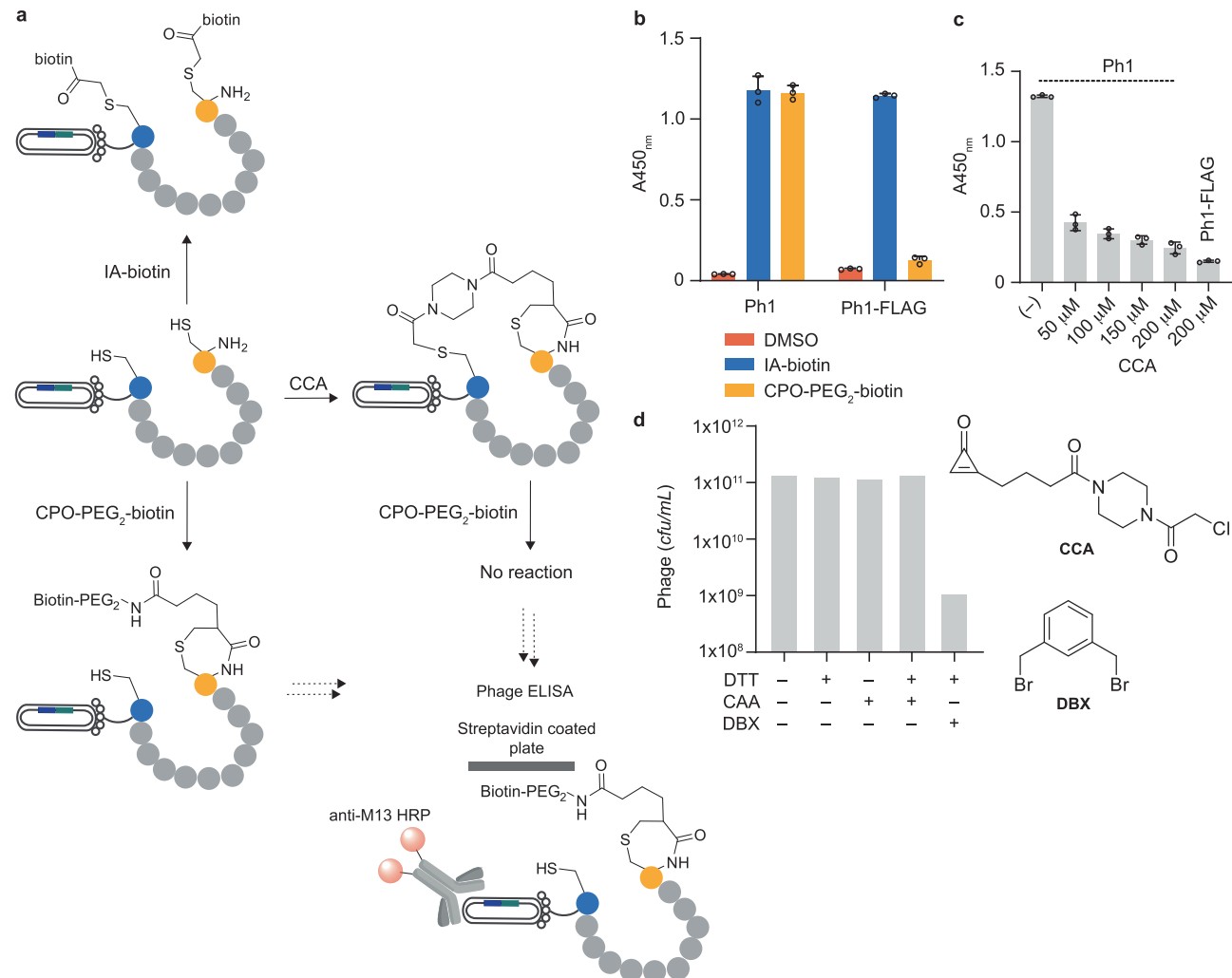

**Fig. 4 | Modification of phage-displayed peptides. a** A schematic representation of the experimental setup used to monitor peptide modification/ cyclization. **b** Using phage ELISA to monitor the CPO modification of phage-displayed peptides with (Ph1) or without (Ph1-FLAG) an N-terminal cysteine residue. Cysteine reactive iodoacetyl-biotin (IA) = positive control. Values are mean ± SD ($n = 3$, technical replicates). **c** Using phage ELISA to monitor the efficiency of CCA-mediated cyclization of phage-displayed peptides with or without an N-terminal cysteine residue.

For Ph1 phage particles, a lower ELISA signal indicates more efficient cyclization. Ph1-FLAG (negative control) does not react with CCA or CPO-PEG$_2$-biotin. Values are mean ± SD ($n = 3$, technical replicates). **d** Titres of phage treated with DTT (1 mM), CCA (150 μM), DTT (1 mM), then CCA (150 μM), or DTT (1 mM) then DBX (150 μM). Before treatment with DBX, the phage was filtered to remove DTT, as in the methods. $Cfu$/mL = [No. of colonies] x [Dilution factor] x [1/Volume plated (mL)].

analogue, whereas no modification of the clipped D1/D2 protein is observed, despite it containing reactive cysteine residues.

## Phage compatibility of CCA peptide cyclization
Following the successful cyclization of vasopressin and model D1/D2 fusion proteins P1–P6, phage particles displaying the peptide CGGSGGC (previously used for generating P2 D1/D2 fusion), proceeded by a FLAG tag (DYKDDDDK) at the N-terminus of pIII, were produced and quantified. Enterokinase cleavage of the FLAG tag was monitored by ELISA and was shown to proceed efficiently (Supplementary Fig. 12).

To optimise conditions for CPO conjugation, enterokinase cleaved phage (Ph1) was treated with 1 mM DTT for 30 min at 37 °C (to reduce the disulfide bond), followed by incubation with 150 μM CPO-PEG$_2$-biotin. The extent of biotinylation was assessed by ELISA on streptavidin using an anti-M13 antibody fused to horseradish peroxidase (HRP) as the detection agent. As a positive control, Ph1 phage particles were treated with iodoacetyl-PEG$_2$-biotin, following a well-established protocol[20]. Comparable positive ELISA readouts were recorded for iodoacetyl-PEG$_2$-biotin and CPO-PEG$_2$-biotin-treated

phage. In contrast, the FLAG-capped phage lacking a free N-terminal cysteine residue yielded little phage capable of binding to streptavidin, following incubation with CPO-PEG$_2$-biotin (Fig. 4b).

Next, the CCA-mediated cyclization of peptides displayed on the Ph1 phage was performed. This consisted of a two-step procedure: (i) reduction of disulfides with DTT followed by (ii) incubation with CCA to cross-link the two cysteines. The extent of CCA-cyclization was assessed by performing a secondary reaction with CPO-PEG$_2$-biotin and capturing biotinylated phage on a streptavidin-coated plate (Fig. 4a). FLAG-capped phage (Ph1-FLAG), which do not react with CCA or CPO-PEG$_2$-biotin, were included as a negative control. A lower ELISA signal equates to more efficient cyclization as CPO-PEG$_2$-biotin capture is blocked by the presence of linker CCA. In the presence of linker CCA, there was a significant drop in phage pulldown, indicating efficient cyclization of the enterokinase-treated phage (Fig. 4c). The effect was greater at a higher CCA concentration, indicating increased cyclization efficiency.

Finally, a titration was performed to quantify the number of infective phages, before and after addition of DTT and linker CCA. Incubating the phage for 30 min with 1 mM DTT and 3 h with 150 μM

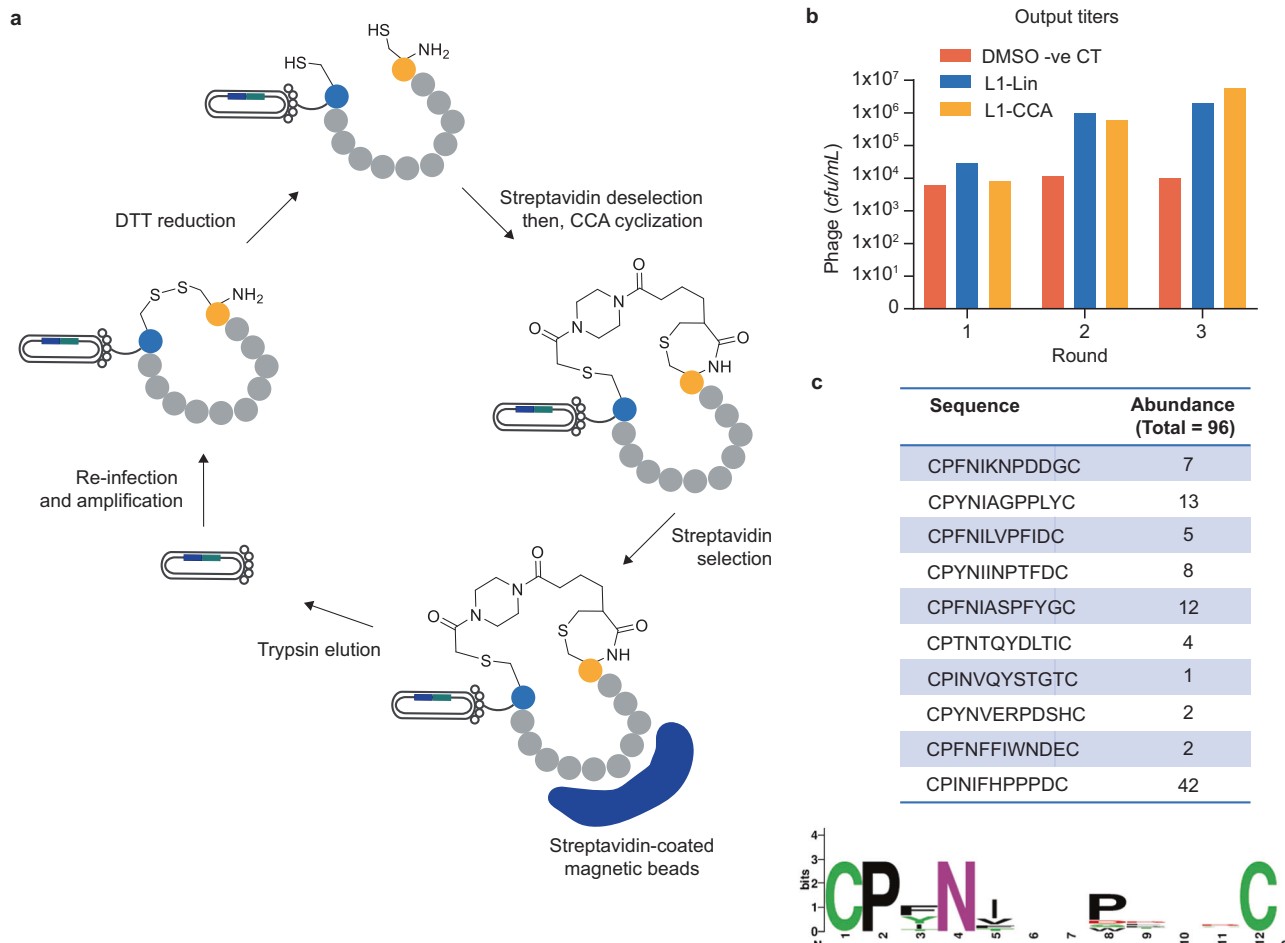

**Fig. 5 | The selection of a novel class of Streptavidin-binding cyclic peptides.** **a** Schematic representation of the phage display setup used to identify streptavidin-binding CCA-cyclized peptides. Reduced L1 phage particles were incubated with streptavidin-coated beads to deplete the pool of linear binders (deselection) prior to CCA cyclization. **b** Phage output titres after rounds 1, 2 and 3 of selection against streptavidin for libraries L1-Lin and L1-CCA. The L1-Lin library was rescued each round with either helper phage or hyper phage. An increase in output population indicates successful enrichment of streptavidin-binding peptides. **c** Individual sequences and consensus sequence (illustrated using WebLogo 3 software v. 7.12) of peptides enriched after three rounds of selection against streptavidin using library L1-CCA.

CCA led to no significant reduction in the number of infective particles, likely due to the high selectivity of CPO reagents for N-terminal cysteine residues. In contrast, a > 100-fold reduction in phage titre was observed upon treatment of the same phage particles for 30 min with 1 mM TCEP and 3 h with 150 μM $m$-dibromoxylene (DBX, used as a model reaction for cyclisation based on the Bicycle® TBMB linker) (Fig. 4d). DBX will react with both TCEP and DTT. Following reduction, the phage must be filtered or precipitated to remove TCEP/DTT prior to cyclization[6,21]. In contrast, CPO conjugation can occur in the presence of large quantities of DTT, so no purification step is required between reduction and cyclization. Therefore, the drop in phage titre can be attributed to both the known toxicity of DBX and the additional purification step. Both factors will lead to a drop in the number of infective phage particles and in turn, lower library size and diversity.

### Construction of CCA-cyclized phage display library and proof-of-concept selections against streptavidin

Following the development of linker CCA and proof of concept peptide cyclization on phage construct Ph1, phage library L1 was designed to display the randomised peptide $CX_{10}C$ (X = any natural amino acid) at the N-terminus of pIII. The library of $4 \times 10^{11}$ transformants after electroporation was constructed using a pC6 phagemid vector and rescued with helper phage to promote low valency display and the selection of high-affinity binders[22]. The FLAG tag technology, although validated for phage, was only applied to the expression and purification of D1/D2-peptide fusions to ensure homogeneity of the product. When generating phage display library L1, the FLAG tag was omitted to streamline the selection procedure. No difficulties were experienced expressing phage-displayed peptides with cysteine in the first position.

Linear, disulfide cyclized and chemically cyclized peptide phage display libraries have all previously been used for the identification of high-affinity streptavidin binders[23,24]. Therefore, streptavidin provided an appropriate target for the validation of CCA-cyclized library L1. Two selection campaigns were performed in parallel: the first on L1 treated with 1 mM DTT (L1-Lin, peptide in linear conformation) and the second on L1 treated with 1 mM DTT then 150 μM CCA (L1-CCA, peptide in CCA cyclized conformation). Prior to CCA cyclization, a deselection step was included to deplete the pool of linear streptavidin binders (Fig. 5a). The depleted library treated with DMSO, instead of linker CCA, was also included as a negative control.

The enrichment of streptavidin-binding peptides was monitored through the sequencing of 48 randomly selected clones after round 3. For both libraries (L1-Lin and L1-CCA), a significant increase in phage output titre was observed after three rounds of selection, and the sequencing results revealed consensus sequences/ dominant motifs (Fig. 5b). As expected, the L1-Lin selection campaign yielded a panel of 'HPQ/M' (well-known streptavidin binding motif)-containing peptides[24].

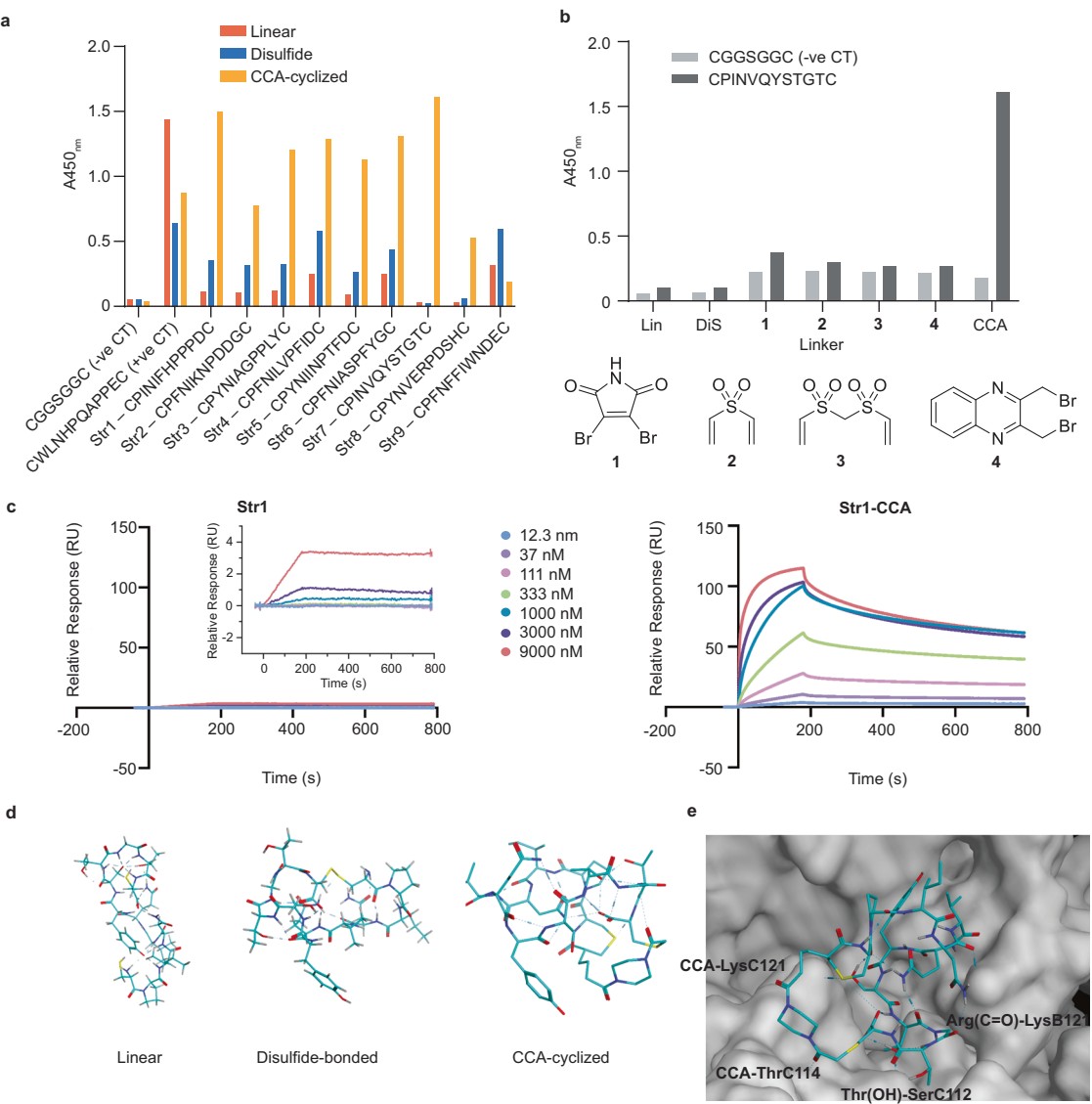

**Fig. 6 | Validation of streptavidin binding via phage ELISA (bound phage detected using anti-M13-HRP) and BLI. a** Binding of nine phage clones (linear, disulfide-bonded or CCA cyclized) to immobilised streptavidin. **b** Binding of phage clone Str7 (linear, disulfide-bonded or cyclized with linker 1, 2, 3, 4 or CCA) to immobilised streptavidin. **c** SPR sensorgrams of peptides Str1 and Str1-CCA interacting with streptavidin. The association time was 180 seconds, and the dissociation time was 600 seconds for each peptide. The curves correspond to one of three independent replicates. Experiments were performed in triplicate. **d** Visualisation of the lowest energy conformations of Str7, a peptide that binds to streptavidin, in its linear, disulfide-bonded, and CCA-cyclized forms (calculated using MOE software version 2020.09). The images show the peptide backbone and linker CCA in stick representation, coloured by atom type. The images provide insight into the structural changes induced by cyclization. **e** Computational docking of CCA-cyclized Str7 (Str7-CCA) bound to streptavidin (calculated using MOE software version 2020.09). Binding site residues involved in peptide interactions are shown in surface representation, coloured in grey. All interactions between Str7-CCA and streptavidin are included.

In contrast, the 'HPQ/M' streptavidin-binding motif was not observed in the L1-CCA output population, confirming the successful depletion of linear streptavidin-binders in the deselection step. Instead, nine unique amino acid sequences were identified with the strong consensus motif CPXNX$_3$PX$_3$C (Fig. 5c). The unique consensus sequence attests that CCA-cyclization provides access to a chemical space largely inaccessible to traditional linear and disulfide-cyclized peptide libraries.

## Validation and characterisation of streptavidin-binding peptides

The binding of all nine enriched phage clones (Str1 – Str9) to streptavidin was tested using phage ELISA under conditions where the peptide was either linear (reduced using DTT), disulfide-bonded (no chemical treatment) or CCA cross-linked (reduced with DTT and reacted with CCA). CCA cyclization was proven essential for the binding of all nine selected peptides (Fig. 6a). The binding of peptide Str7, which, when displayed on phage only binds to streptavidin when CCA cyclized, was then tested in the context of alternative cross-linkers. Using phage ELISA, no binding was observed when the CCA linker was replaced with linkers 1, 2, 3 or 4 (Fig. 6b). We hypothesise that CCA holds Str7 in a specific conformation that is required for binding. Upon replacement of CCA with a different cysteine reactive linker, the conformation is likely altered and thus, binding disrupted. No binding was observed between the CCA-cyclized negative control phage construct (CGGSGGC) and streptavidin, indicating that interactions between linker CCA and streptavidin do not contribute towards the binding of Str7-CCA.

High-affinity binding of synthetic peptide Str1-CCA, the most abundant peptide from phage display selections, was confirmed using surface plasmon resonance ($K_D = 0.34 \mu M$, Supplementary Fig. 18). The

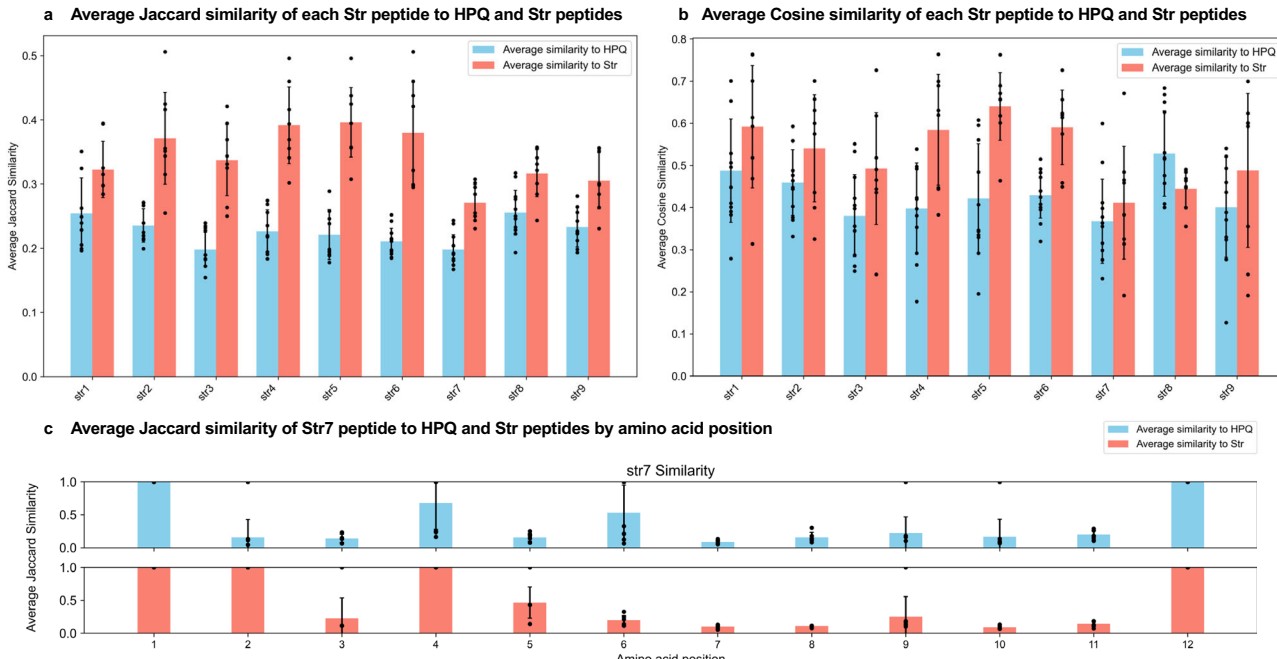

**Fig. 7 | Three different approaches to evaluate the chemical diversity and dissimilarity between CCA cyclized peptides (Str1-9, sequences obtained from L1-CCA library selections) and linear peptides exhibiting the HPQ pattern (HPQ1-12, sequences obtained from L1-Lin library selections). a** Jaccard similarity (1 – distance) between MinHash vectors of MAP4 fingerprints. **b** Cosine similarity between amino acid text sequences using TfidfVectorizer. **c** Jaccard similarity between MinHash vectors of MAP4 fingerprints for each position in the peptide sequence.

presence of linker CCA proved essential for binding, with no SPR signal observed for synthetic peptide Str1 as a disulphide-bonded entity (Fig. 6c).

To gain structural insight into the peptide-streptavidin binding, computational studies were performed. To begin, Str7 was visualised in its linear, disulfide-bonded and CCA-cyclized forms. The conformation of the peptide was found to change substantially upon removal of linker CCA, which likely explains the change in binding affinity observed in the ELISA and BLI experiments (Fig. 6d). Next, in silico docking of CCA-cyclized Str7 to streptavidin was performed to explore potential binding sites and interactions (Fig. 6e). The results revealed direct interactions between Str7-CCA and streptavidin and a binding site distinct from that of biotin and other known peptide-based streptavidin binders (e.g., strep tag and strep tag II)[25,26]. These computational studies suggest that Str7 binding is highly dependent on the specific conformation induced by CCA-mediated cyclization. The identified binding site is a target for streptavidin binding, which could have potential applications in various biotechnological fields. Further experimental studies would be required to confirm the binding site/ mechanism.

## Cheminformatic analysis to quantify the divergence between our reported peptide sequences and known streptavidin binders

To understand and evaluate the chemical diversity and dissimilarity between CCA cyclized peptides (Str1-9, sequences obtained from L1-CCA library selection) and linear peptides exhibiting the HPQ pattern (HPQ1-12, sequences obtained from L1-Lin library selection), we employed chemoinformatic analysis using three distinct methods. We converted the complete peptide sequences into SMILES strings and then calculated the MinHashed Atom-Pair Fingerprint (MAP4). MAP4 encodes atom–bond pairs[27] and was chosen for its effectiveness in mapping the chemical space of a wide array of molecules. Then, the Jaccard distance was calculated to provide a similarity measure between two MinHash vectors (1 for identical, 0 for highly dissimilar). Alternatively, each amino acid was converted into SMILES strings, creating a dictionary for each of the twelve amino acid positions in

each peptide. This was followed by MAP4 fingerprinting and Jaccard distance calculation for each position leading to positional similarity. Finally, we counted the occurrence of each pepamino acid by applying natural language processing methods and then calculated the cosine similarity to compare text similarities of the written peptide sequences.

We calculated the average similarity values for each Str peptide (Str1-9) against HPQ1-12 and other streptavidin-binding peptides (excluding self-comparisons) following the three methods. The results are presented in Fig. 7. The Jaccard similarity of the entire peptide sequences (Fig. 7a) indicates a lower average similarity between Str and HPQ peptides. Str7 showed particularly low similarity, highlighting its unique amino acid sequence even within the Str family. This trend is also observed for the cosine similarity (Fig. 7b) but with a higher variability. Positional analysis of Str7 (Fig. 7c) revealed few consistent features between Str and HPQ peptides. Whilst the fourth position, often occupied by asparagine, is similar in both peptide groups, dissimilarity is observed in the second position, where Str peptides consistently feature proline, contrasting with the diverse residues found in HPQ peptides. In the other positions, we found that Str7 has a low similarity for both HPQ and Str peptides.

## Phage library selections against the therapeutically relevant target αvβ3 integrin

The general applicability of our CCA-cyclized peptide phage display library was demonstrated by performing selections against integrin αvβ3. Integrins are a family of cell surface receptors that play a key role in cell adhesion, migration, proliferation, and differentiation. The integrin αvβ3 is a particularly important member of this family as it is expressed on the surface of many different cell types, including endothelial cells, osteoclasts, and tumour cells. It plays a critical role in angiogenesis, bone resorption, and tumour growth and metastasis[13,28].

After just two rounds of selection, a significantly increased output population was observed and a dominant 'RGD' motif was identified in

the sequencing data. To drive for high-affinity binders, the concentration of antigen was reduced to 1 nM for round 3. The phage output titre remained high (Supplementary Fig. 15) and sequencing revealed a diverse array of 'RGD' containing peptides. Notably, many of these peptides have an 'ionic zipper format' – i.e., a central RGD motif surrounded by negatively charged amino acids on one side and positively charged amino acids on the other. Integrin αvβ3 recognises the RGD motif as a ligand-binding site[13]. Therefore, this 'ionic zipper' format presumably pushes the RGD motif as far away from pIII as possible, limiting steric hindrance and maximising binding.

Eight dominant clones containing the RGD motif flanked by different sequences were selected for further analysis. The specific binding of the CCA-cyclized peptides was confirmed using phage ELISA (Supplementary Fig. 16). Interestingly, all tested peptides bind to αvβ3 in the absence of linker CCA. In other words, our αvβ3 binding peptides appear to benefit less from CCA cyclization than our streptavidin binding peptides, potentially due to the conformational rigidity imparted by the 'ionic zipper' format. Peptides av1, av2 and av3 were also expressed as D1/D2 fusions, cyclized with linker CCA and low nM ($k_D$ = 7–14 nM) binding to αvβ3 was confirmed using BLI (Supplementary Fig. 17).

Overall, the success of the αvβ3 selection campaign demonstrates the potential of CCA-cyclized peptide phage display libraries for use against therapeutically relevant targets. The ability to identify stable cyclic peptides that bind with high affinity to important cellular receptors such as αvβ3 presents a significant advancement in the development of targeted therapeutics, diagnostics and drug delivery. For example, cells that express αvβ3 could be selectively labelled in a non-invasive manner by conjugating fluorescent dyes or radioactive isotopes to αvβ3-targeting cyclic peptides for diagnostic purposes. Alternatively, αvβ3-targeting cyclic peptides could be used for targeted delivery of drugs to specific cells or tissues that express αvβ3, reducing the toxicity and increasing the efficacy of the drug.

## Discussion

We have established an efficient method for the reduction and cyclization of phage-displayed peptides using DTT and a CPO-based proximity-driven cyclization linker (linker CCA). The site-specificity of linker CCA for N-terminal cysteine modification and subsequent proximity-driven cyclization was exemplified on peptides, D1/D2-peptide fusion proteins and individual phage constructs. CPO chemistry is compatible with large quantities of DTT, so unlike other cysteine-reactive cross-linkers, no purification step is required between reduction and cyclization. Furthermore, linker CCA does not affect the infectivity of phage particles, nor does it require the use of disulfide-free pIII. Therefore, this one-pot reduction and cyclization method facilitates the production of large and highly diverse cyclic peptide phage display libraries.

Using our cyclization strategy, we were able to construct a CCA-cyclized peptide phage display library, which, when subjected to selections against streptavidin yielded a novel class of binders. The uniqueness of our cyclization method has facilitated the discovery of streptavidin binders beyond the conventional HPQ motif for the first time through phage display. Furthermore, these binders are solely dependent on the presence of CCA crosslinker and non-functional as linear or disulphide-bonded entities. Further selections against αvβ3 yielded binders with low nM affinity, confirming that our cyclization approach can be applied to produce cyclic peptide binders against therapeutically relevant targets.

One potential limitation of our cyclization approach is that CCA-cyclized peptides can adopt four different conformations, which can complicate the identification of true binders during the phage display selection process. However, a mixture of stereoisomers also increases the theoretical diversity of the randomised library and further expands the chemical space of phage display.

Future work will focus on building libraries of different sizes using CCA linkers of different length and rigidity. Further selections will be performed against a variety of targets, with a focus on proteins that require a reducing environment to maintain their active form (correct oxidation state or conformation). In contrast to traditional disulfide-bonded peptide libraries, CCA-cyclized libraries provide a promising alternative format for reducing environment selections on account of their stability in the presence of TCEP and DTT.

## Methods

### LC-MS methods

**Proteins.** LC-MS analysis of protein samples was executed using an SQ Detector 2 mass spectrometer connected to an Acquity UPLC system with an Axquity UPLC BEH300 C4 column (1.7 μm, 2.1 mm × 50 mm). The eluents used were water with 0.1% (v/v) formic acid (solvent A) and 71% (v/v) acetonitrile in water with 0.075% (v/v) formic acid (solvent B). The flow rate was kept constant at 0.2 mL/min and the gradient was programmed as follows: from a starting point of a 72:28 ratio of A:B, the gradient moved to 71.2% B over 12 min, then moved to 100% B over 1 min, remained at 100% B for 3 min, then moved back to the starting ratio of 72:28 A:B (72:28) over 0.5 min, and remained at this ratio for 3 min. The electrospray source was set at 3.0 kV capillary and 30 V cone voltage. Nitrogen was used as the desolvation gas at 800 L/h. Mass spectra were reconstructed from the measured ion series using the MaxEnt algorithm within MassLynx software (Waters).

**Peptides.** LC-MS analysis was performed for peptides as described above for proteins. The only significant changes were the column type and solvent gradient. An Acquity UPLC BEH300 C18 column (1.7 μm, 2.1 mm × 50 mm) was used and the gradient (with the same eluent system) was as follows: 2 min at 100% solvent A, a gradient moving steadily to reach 100% solvent B at 11 min, remaining at 100% solvent B until 16 min, and then moving back to 100% solvent A at 16.1 min until the end of the run at 20 min.

### Peptide reactions

All peptides were purchased from GenScript (Supplementary Information). Peptides (100 μM) were reduced with DTT (200 μM) for 1 h at room temperature and then incubated with the indicated compound (200 μM) in PBS (pH 7.4) for 2 h at room temperature. The reaction was monitored and analysed by LC-MS.

### D1/D2-peptide fusions

**Plasmid construction.** Template plasmid pCantab6 D1/D2 (10 μg, supplied by AstraZeneca) was mixed with 10x CutSmart® buffer (5 μL), HindIII-HF®/NotI-HF® restriction enzymes (2.5 μL each) and sterile water (total reaction volume = 50 μL). Samples were incubated for 2 h at 37 °C then analysed via agarose gel electrophoresis to confirm complete digestion. The digested plasmid was excised from the gel, purified with a QIAquick® Gel extraction Kit and quantified using a Lunatic UV/Vis spectrometer (λ = 260 nm, Unchained Labs).

Hybrid plasmids P1-FLAG, P1-TEV, P1-Xa, P2-FLAG, P3-FLAG and P4-FLAG were constructed via Gibson Assembly® following the standard NEB protocol and using the strings specified in the Supplementary Information, transformed into chemically competent BL21 (DE3) *E. coli* and plated O/N at 37 °C on 2xTYAG agar. The next day, plasmid DNA was isolated from individual colonies (5 per construct) using a Plasmid Plus Mini Kit (Qiagen) and sequenced (Source BioScience).

Hybrid plasmids P5-FLAG and P6 were kindly gifted by Dr Carole Urbache (AstraZeneca).

**Protein production and purification.** 2xTYAG media (10 mL) was inoculated with a single transformed BL21 (DE3) colony and incubated O/N at 30 °C and 280 RPM. The starter culture was diluted (50x in 2xTYAG) and incubated at 25 °C and 280 RPM until $OD_{600nm}$ = 0.6.

Expression was induced by the addition of IPTG (50 µL, 1.0 M). After 16 h at 25 °C and 250 RPM, the cultures were centrifuged for 20 mins at 4 °C and 2500 × $g$. The cell pellet was resuspended in TES buffer (1 mL), incubated at room temperature for 15 min, added to cold 1:5 TES buffer: sterile water (1.5 mL) and left on ice for 30 min. The periplasmic fraction was clarified via centrifugation at 4 °C and 2500 × $g$ for 30 min. Fusion proteins P1-FLAG, P1-TEV, P1-Xa and P6 were purified using His SpinTrap™ columns (GE Healthcare). Fusion proteins P2-FLAG, P3-FLAG, P4-FLAG and P5-FLAG were purified using Pierce™ Anti-DYKDDDDK Affinity Resin (Thermo Fisher Scientific). Protein concentration was quantified using a Lunatic UV/Vis spectrometer ($\lambda$ = 280 nm, Unchained Labs).

**SDS-PAGE.** Purified protein (2 µg) was mixed with 4x Novex™ Sharp Pre-stained Protein Standard (2.5 µL, Thermo Fisher Scientific), TCEP (1 µL, 500 mM) and sterile water (total reaction volume = 10 µL). The samples were heated at 96 °C for 2 min, then loaded into the wells of a NuPAGE 4-12% Bis-Tris gel (Invitrogen) alongside a SeeBlue™ Plus2 Pre-stained Protein Standard (10 µL, Thermo Fisher Scientific). Electrophoresis was run with 1xNuPAGE™ MES SDS Running Buffer (Thermo Fisher Scientific) at 200 V for 30 min. Protein bands were visualised using InstantBlue™ Coomassie Protein Stain (Abcam) and photographed using a Chemidoc™ MP imaging system (Biorad) fitted with a white tray (coomassie blue protein gel setting).

**CCA cyclization of D1/D2-peptide fusions.** FLAG-capped proteins (25 µg) were combined with reaction buffer (20 mM Tris-HCl, 50 mM NaCl, 2 mM CaCl$_2$, pH 8.0) to a total reaction volume of 20 µL. Enterokinase light chain enzyme (NEB) was added (1 µL) and the reaction was incubated for 16 h at 25 °C. Samples were then buffer exchanged into PBS (pH 7.4) using Amicon Ultra 0.5 spin columns following the manufacturer's instructions.

Enterokinase-cleaved proteins (10 µM) were reduced with DTT (1 mM) in PBS (pH 7.4) for 30 min at 37 °C in a total volume of 95 µL. Linker CCA (250 µM) in DMSO (5% total volume) was then added and the reaction was left to incubate for 2 h at room temperature. All reactions were monitored and analysed by LC-MS.

### Phage production, cyclization and infectivity studies
**Production of Ph1-FLAG phage particles.** Plasmid F2-FLAG (section S0) was transformed into a chemically competent Mix and Go! TG1 *E. coli* (Zymo) and plated O/N at 37 °C on 2xTYAG agar. The next day, 2xTYAG (25 mL) was inoculated with a single transformed TG1 colony and the culture was incubated at 37 °C and 280 RPM until OD$_{600nm}$ = 0.5–1.0. Helper phage was added (2.5 µL) and the mixture was incubated at 37 °C and 150 RPM for 1 h. The cultures were then centrifuged at 3500 RPM for 15 min at room temperature. The supernatant was decanted, and the cell pellet was resuspended in 2xTYAK media (25 mL).

Following O/N incubation at 25 °C and 280 RPM, the culture was centrifuged at 8000 RPM and 4 °C for 20 min. The supernatant was collected in a pre-chilled bottle, and phage particles were precipitated by adding 2.5 M sodium chloride with 20% (v/v) PEG (7.5 mL) and storing it on ice for 2 h.

The supernatant was transferred to a clean centrifuge pot and the phage particles were pelleted by centrifugation at 8000 RPM and 4 °C for 20 min. The supernatant was decanted and the phage pellet was resuspended in PBS (2 mL). Finally, the phage stock was centrifuged at 11,000 RPM and 4 °C for 10 min and the supernatant was stored at 4 °C.

**Modification of Ph1-FLAG and Ph1 phage particles.** Ph1-FLAG phage particles (~ 10$^{11}$) were combined with reaction buffer (20 mM Tris-HCl, 50 mM NaCl, 2 mM CaCl2, pH 8.0) to a total reaction volume of 100 µL. Enterokinase light chain enzyme (NEB) was added (2 µL) and the reaction was incubated for 16 h at 25 °C. Enterokinase-cleaved Ph1-FLAG (Ph1) phage particles were buffer exchanged into PBS (pH 7.4)

using Zeba™ Spin Desalting Columns (7 K MWCO, 0.5 mL) following the manufacturer's instructions.

Ph1-FLAG and Ph1 phage particles (~ 10$^{11}$) were reduced with DTT (1 mM) for 30 min at 37 °C in a total volume of 100 µL. Linker CCA (final concentration = 150 µM) in DMSO (5% total volume) was added and the reaction was left to incubate for 2 h at room temperature. CPO-PEG$_2$-biotin (final concentration = 150 µM) in DMSO (5% total volume) was then added and the reaction was left to incubate for a further 2 h at room temperature. All reactions were monitored and analysed by phage ELISA.

Positive control: Ph1-FLAG and Ph1 phage particles (~ 10$^{11}$) was modified with iodoacetyl-PEG$_2$-biotin (final concentration = 1 mM) following a literature procedure[2].

Negative control: Ph1-FLAG and Ph1 phage particles (~ 10$^{11}$) were treated with DMSO (5% total volume) instead of linker CCA then CPO-PEG$_2$-biotin as described above.

**Phage ELISA.** <u>Enterokinase cleavage ELISA:</u> Biotinylated antigen = Biotinylated Anti-His Tag Antibody (Acro Biosystems), assay buffer = PBS, blocking buffer = 3% (w/v) mPBS. <u>CCA cyclization ELISA:</u> Biotinylated antigen = none required, assay buffer = N/A, blocking buffer = 3% (w/v) mPBS.

All incubation steps were performed at room temperature unless otherwise stated. A Pierce™ streptavidin-coated 96 well plate (Thermo Fisher Scientific) was treated O/N at 4 °C with biotinylated antigen (50 µL/well, 10 ng/µL in assay buffer). The next day, the plate was washed x3 with PBS to remove unbound antigen and incubated for 1 h with blocking buffer (300 µL/ well). At the same time, phage cultures (100 µL, ~ 10$^{10}$) were incubated for 1 h with blocking buffer (400 µL).

The 96-well plate was washed x3 with PBS and subsequently incubated with blocked phage (50 µL/ well) for 1 h. The plate was washed x3 with PBST, and bound phage was detected via the addition of anti-M13 HRP conjugated secondary antibody (50 µL/ well, 1/5000 dilution in blocking buffer, Abcam). After 1 h, the plate was washed x3 with PBST. HRP was developed using TMB substrate (50 µL/ well, Thermo Fisher Scientific), quenched after 5 min with H$_2$SO$_4$ (50 µL/ well, 0.5 M) and visualised at $\lambda$ = 280 nm using an EnVision Microplate Reader (PerkinElmer). Data was processed using SoftMax® Pro Software v7.

**Infectivity studies.** <u>CCA:</u> Ph1 phage particles (~ 10$^{11}$) were reduced with DTT (1 mM) for 30 min at 37 °C in a total volume of 95 µL. Linker CCA (final concentration = 150 µM) in DMSO (5% total volume) was added and the reaction was left to incubate for 2 h at room temperature. <u>DBX:</u> Ph1 phage particles (~ 10$^{11}$) were reduced with TCEP (1 mM) for 30 min at 37 °C in a total volume of 95 µL. Reduced Ph1 phage particles were buffer exchanged into NH$_4$HCO$_3$ (20 mM, pH 8.0) using Zeba™ Spin Desalting Columns (7 K MWCO, 0.5 mL) following the manufacturer's instructions. Linker DBX (final concentration = 150 µM) in DMSO (5% total volume) was added and the reaction was left to incubate for 2 h at room temperature.

For each sample, two 100-fold dilutions in 2xTY were prepared. An aliquot (10 µL) of the 10$^{-4}$ dilution sample was added to TG1 *E. coli* (990 µL) grown to OD$_{600nm}$ = 0.6. After 1 h of infection at 37 °C and 150 RPM, four further 10-fold dilutions in 2xTY were prepared (10$^{-7}$ – 10$^{-10}$). Aliquots (100 µL) of dilution samples 10$^{-7}$ – 10$^{-10}$ were plated O/N on 2xTYAG agar at 37 °C. The next day, phage titres were calculated according to the equation.

Cfu/mL = [No. of colonies] x [Dilution factor] x [1/Volume plated (mL)].

### Library construction and cyclization
#### Kunkel mutagenesis
**Preparation of dU-ssDNA.** Stop template ST1, kindly provided by Dr Carole Urbache (AstraZeneca), was transformed into chemically

competent CJ236 *E. coli* (Supplementary Information). 2xTYAGC (5 mL) inoculated with a single transformed CJ236 colony was incubated at 37 °C and 300 RPM until $OD_{600nm}$ = 0.8 – 1. Wild-type M13KO7 helper phage (provided by AstraZeneca) was added to a multiplicity of infection of 10 (1 μL, $3 \times 10^{13}$ phage/mL stock) and the mixture was incubated without shaking for 10 min at 37 °C. An aliquot (1 mL) of the culture was diluted in 2xTYAK (30 mL) supplemented with uridine (0.25 μg/mL).

Following O/N incubation at 37 °C and 300 RPM, the cells were harvested by centrifugation for 10 min at 2 °C and 15,000 RPM. The supernatant was transferred to a fresh tube and 2.5 M sodium chloride with 20% (v/v) PEG (9 mL) was added. Phage particles were collected by incubating at room temperature for 10 min and centrifugation for 10 min at 2 °C and 10,000 RPM. The supernatant was decanted and the phage pellet was resuspended in PBS (0.5 mL) and centrifuged for 5 min at room temperature and 13,000 RPM. The supernatant was transferred to a clean Eppendorf and the dU-ssDNA template was purified using an E.Z.N.A.® M13 DNA Mini Kit.

**Phosphorylation of mutagenic oligonucleotides.** Mutagenic oligonucleotide O1 (0.7 μg) was mixed with 10x TM buffer (2 μL), ATP (2 μL, 10 mM), DTT (1 μL, 100 mM) and sterile water (total reaction volume = 18 μL). T4 polynucleotide kinase (2 μL, 10 U/μL) was then added and the mixture was incubated for 1 h at 37 °C.

**Annealing of mutagenic oligonucleotide to template at a ratio of 3:1.** To the phosphorylated oligo (20 μL) was added ST1 dU-ssDNA (20 ug), 10x TM buffer (25 μL) and sterile water (total reaction volume = 250 μL). The mixture was split equally between two PCR tubes and incubated at 90 °C for 2 min, 50 °C for 3 min and 20 °C for 5 min.

**Mutagenesis reaction.** To the annealed oligo/ template (250 μL) was added ATP (10 μL, 10 mM), dNTPs (10 μL, 25 mM), DTT (15 μL, 100 mM), T4 DNA ligase (5 μL, 6 U/μL) and T7 DNA polymerase (3 μL, 10 U/μL). The mixture was split evenly between two reaction wells per library (21.5 μL/well) and incubated for 3 h at 20 °C.

The reaction product was affinity-purified into water (50 μL) using a Roche High Pure DNA purification kit and transformed into electrocompetent HB2151 *E. coli*. For this, an aliquot of cells (1.6 mL) was added to an Eppendorf alongside the mutated plasmid (500 ng). The cell/ DNA mixture was equally portioned into four prechilled 0.2 mm Gene Pulser/ MicroPulser Electroporation Cuvettes (Invitrogen). Each electroporation pulse was executed at 2500 V, 15 μF, and 335 R. Immediately after electroporation, 2xTYG (1 mL, prewarmed to 37 °C) was added to each cuvette and the cell suspension was transferred to a 50 mL Falcon tube (one per library), rinsing with 2xTYG (1 mL). Following 1 h of incubation at 37 °C and 150 RPM, the cells were pelleted by centrifugation for 10 min at room temperature and 3200 RPM, resuspended in 2xTYG (1 mL) before plating O/N on a 2xTYAG bioassay plate at 37 °C.

The next day, colonies were picked and prepared for sequencing via colony PCR. The library was scraped from the bioassay plate into 2xTY media (5 mL) mixed with 50% (v/v) glycerol (2.5 mL), portioned into 500 μL aliquots and stored at − 80 °C.

**Phage library rescue.** L1 HB2151 glycerol backup stock was added to 2xTYAG media (400 mL) to an $OD_{600nm}$ ~ 0.1. The bacterial culture was incubated at 37 °C and 280 RPM until $OD_{600nm}$ = 0.5–1.0. Helper phage was added (80 μL) and the mixture was incubated at 37 °C and 150 RPM for 1 h. The cultures were pelleted by centrifugation for 15 min at room temperature and 3500 RPM and resuspended in 2xTYAK media (400 mL).

Following O/N incubation at 25 °C and 280 RPM, the culture was centrifuged at 8000 RPM and 4 °C for 20 min. The supernatant was collected in a pre-chilled bottle, and phage particles were precipitated

by adding 2.5 M sodium chloride with 20% (v/v) PEG (120 mL) and stored on ice for 1 h.

The supernatant was transferred to a clean centrifuge pot and the phage particles were pelleted by centrifugation at 8000 RPM and 4 °C for 20 min. The supernatant was decanted, the phage pellet was resuspended in TE buffer (10 mL) and 2.5 M sodium chloride with 20% (v/v) PEG (3 mL) was added.

Following a 1 h precipitation on ice, the phage stock was centrifuged at 11,000 RPM and 4 °C for 15 min. The supernatant was decanted and the phage pellet was resuspended in TE buffer (3 mL). Finally, the phage stock was centrifuged at 11,000 RPM and 4 °C for 10 min and the supernatant was stored at 4 °C.

### Phage display selections
#### Streptavidin selection
**Negative selection and CCA cyclization.** Streptavidin-coated magnetic Dynabeads™ (50 μL/ selection, prewashed with 1 mL PBS) were blocked with 3% (w/v) mPBS (1 mL) for 1 h on a rotary mixer at room temperature and 20 RPM. L1 phage particles (~ $10^{11}$) were reduced with DTT (1 mM) for 30 min at 37 °C in PBS (total reaction volume = 100 μL). The beads were pelleted on a magnetic stand and reduced L1 phage particles (~ $10^{11}$) were added. After 1 h on a rotary mixer at room temperature and 20 RPM, the beads were pelleted and the supernatant was transferred to a fresh Eppendorf. Linker CCA (final concentration = 150 μM) in DMSO (5% total volume) was added and the reaction was left to incubate for 2 h at room temperature.

**Positive selection.** L1-Red: L1 phage particles (~ $10^{11}$) were reduced with DTT (1 mM) for 30 min at 37 °C in PBS (total reaction volume = 100 μL).

L1-CCA: Output phage from the negative streptavidin selection above.

L1-Red/ L1-CCA input phage (~ $10^{11}$) and streptavidin-coated magnetic Dynabeads™ (50 μL/ selection, prewashed with 1 mL PBS) were incubated with 3% (w/v) mPBS (volume made up to 500 μL for phage, 1 mL for beads) for 1 h on a rotary mixer at room temperature and 20 RPM. The beads were pelleted on a magnetic stand and resuspended in 3% (w/v) mPBS (100 μL). An aliquot of the blocked beads (50 μL) was added to the blocked phage. The mixture was incubated for 1 h on a rotary mixer at room temperature and 20 RPM.

Using a Kingfisher mL apparatus, the beads were washed x5 with PBST (1 mL) and eluted into trypsin solution (10 μg/ mL trypsin in 0.1 M pH 7.0 sodium phosphate buffer, 200 μL). Trypsin digest was performed for 30 min at 37 °C and 600 RPM, the beads were pelleted and the supernatant was added to $OD_{600nm}$ = 0.6 TG1 *E. coli* (800 μL). After 1 h at 37 °C and 250 RPM, output titres were performed and the remaining cells were plated O/N on a 2xTYAG bioassay plate at 37 °C. The next day, the cells were harvested from the plate, as described below.

#### αvβ3 selection
**Negative selection and CCA cyclization.** Assay buffer = HEPES (pH 7.0, 50 mM), NaCl (150 mM), MnCl (0.1 mM)

Blocking buffer = HEPES (pH 7.0, 50 mM), NaCl (150 mM), MnCl (0.1 mM) with 2% (v/v) BSA.

Streptavidin-coated magnetic Dynabeads™ (50 μL/ selection, prewashed with 1 mL PBS) were blocked with 3% (w/v) mPBS (1 mL) for 1 h on a rotary mixer at room temperature and 20 RPM. L1 phage particles (~ $10^{11}$) were reduced with DTT (1 mM) for 30 min at 37 °C in assay buffer (total reaction volume = 100 μL). Biotinylated αvβ3 (50 nM, Acro Biosystems) was added and the mixture was incubated for 1 h on a rotary mixer at room temperature and 20 RPM. The beads were pelleted on a magnetic stand and the reduced L1 phage particle/ biotinylated αvβ3 mixture was added. After 1 h on a rotary mixer at

room temperature and 20 RPM, the beads were pelleted and the supernatant was transferred to a fresh Eppendorf.

The supernatant was buffer exchanged into PBS (pH 7.4) using Zeba™ Spin Desalting Columns (7 K MWCO, 0.5 mL) following the manufacturer's instructions. Linker CCA (final concentration = 150 μM) in DMSO (5% total volume) was added and the reaction was left to incubate for 2 h at room temperature.

**Positive selection.** L1-Red: L1 phage particles ($\sim 10^{11}$) were reduced with DTT (1 mM) for 30 min at 37 °C in PBS (total reaction volume = 100 μL).

L1-Dis: L1 phage particles with no further modification.

L1-CCA: Output phage from the negative streptavidin selection above.

L1-Red/ L1-DiS/ L1-CCA input phage ($\sim 10^{11}$) and streptavidin-coated magnetic Dynabeads™ (50 μL/ selection, prewashed with 1 mL PBS) were incubated with blocking buffer (volume made up to 500 μL for phage, 1 mL for beads) for 1 h on a rotary mixer at room temperature and 20 RPM. The beads were pelleted on a magnetic stand and resuspended in a blocking buffer (100 μL). To deselect against streptavidin, an aliquot of the blocked beads (50 μL) was added to the blocked phage. After 1 h on a rotary mixer at room temperature and 20 RPM, the beads were pelleted and the supernatant was transferred to a fresh Eppendorf. Biotinylated antigen was added at the concentration required. After 1 h rotating on an end-over-end rotor, the remaining aliquot of blocked beads (50 μL) was added and the mixture was equilibrated for 5 min on an orbital shaker at 37 °C and 300 RPM.

Using a Kingfisher mL apparatus, the beads were washed x5 with PBST (1 mL) and eluted into trypsin solution (10 μg/ mL trypsin in 0.1 M pH 7.0 sodium phosphate buffer, 200 μL). Trypsin digest was performed for 30 min at 37 °C and 600 RPM, the beads were pelleted and the supernatant was added to $OD_{600nm}$ = 0.6 TG1 *E. coli* (800 μL). After 1 h at 37 °C and 250 RPM, output titres were performed and the remaining cells were plated O/N on a 2xTYAG bioassay plate at 37 °C. The next day, the cells were harvested from the plate, as described below.

**Selection rescue.** Infected bacteria were scraped from the 2xTYAG bioassay plate with glycerol-medium (10 mL, 2xTY media mixed with 50% v/v glycerol in a 2:1 ratio) using a disposable plastic spreader. The cell suspension was transferred into a 50 mL Falcon tube and placed on an end-over-end rotor for 10 min to fully resuspend the bacterial plate scrape. An aliquot of the plate scrape (1 mL) was stored at −80 °C as a backup stock.

2xTYAG media (25 mL) was inoculated with a sufficient quantity of the bacterial plate scrape to reach $OD_{600nm}$ = 0.1. The bacterial culture was incubated at 37 °C and 280 RPM until $OD_{600nm}$ = 0.5-1.0. M13K07$^{trp}$ helper phage (2.5 μL, $3 \times 10^{13}$ cfu/mL stock) were added and the culture was incubated for 1 h at 37 °C and 150 RPM. The cells were centrifuged at 3200 xg for 10 min and resuspended in 2xTYAK media (25 mL).

Following O/N incubation at 25 °C and 280 RPM, the culture was centrifuged at 8000 RPM and 4 °C for 20 min. The supernatant was collected in a pre-chilled bottle, and phage particles were precipitated by adding 2.5 M sodium chloride with 20% (v/v) PEG (7.5 mL) and storing it on ice for 2 h.

The supernatant was transferred to a clean centrifuge pot and the phage particles were pelleted by centrifugation at 8000 RPM and 4 °C for 20 min. The supernatant was decanted and the phage pellet was resuspended in PBS (2 mL). Finally, the phage stock was centrifuged at 11,000 RPM and 4 °C for 10 min and the supernatant was stored at 4 °C.

**Input and output titres.** 2xTY media (50 mL) was inoculated with a fresh single TG1 colony and incubated at 37 °C and 280 RPM until $OD_{600nm}$ = 0.6.

### Input titres
A serial dilution of the input phage stock was performed as follows:

(1) 10 μL input phage + 990 μL 2xTY media(1:100)$10^2$
(2) 10 μL (1) + 990 μL 2xTY media(1:100)$10^4$
(3) 10 μL (2) + 990 μL $OD_{600nm}$ = 0.6 TG1(1:100)$10^6$
 Sample (3) was incubated for 1 h at 37 deg and 150 RPM then diluted as below:
(4) 100 μL (3) + 900 μL 2xTY media(1:10)$10^7$
(5) 100 μL (4) + 900 μL 2xTY media(1:10)$10^8$
(6) 100 μL (5) + 900 μL 2xTY media(1:10)$10^9$
(7) 100 μL (6) + 900 μL 2xTY media(1:10)$10^{10}$

An aliquot of dilution samples $10^7 - 10^9$ (100 μL) was plated O/N on 2xTYAG agar at 37 deg. The next day, input titres were calculated according to the equation:

Cfu/mL = [No. of colonies] x [Dilution factor] x [1/Volume plated (mL)].

### Output titres
A serial dilution of the output phage stock was performed as follows:

(1) 10 μL output phage + 90 μL 2xTY media(1:10)$10^1$
(2) 10 μL output phage + 990 μL 2xTY media(1:100)$10^2$
(3) 100 μL (2) + 900 μL 2xTY media (1:10)$10^3$
(4) 100 μL (3) + 900 μL 2xTY media(1:10)$10^4$

An aliquot of dilution samples $10^1 - 10^4$ (100 μL) was plated O/N on 2xTYAG agar at 37 deg. The next day, output titres were calculated according to the equation:

*Cfu/mL* = [No. of colonies] x [Dilution factor] x [1/Volume plated (mL)].

### Peptide hit validation
**Phage ELISA.** Individual phage constructs were rescued, precipitated, reduced with DTT and cyclized with linker CCA as described previously (section 0).

**Streptavidin-binding ELISA:.** Biotinylated antigen = none required, assay buffer = N/A, blocking buffer = 3% (w/v) mPBS.

**αvβ3-binding ELISA:.** Biotinylated antigen = biotinylated αvβ3 (Acro Biosystems), assay buffer = HEPES (pH 7.0, 50 mM), NaCl (150 mM), MnCl (0.1 mM), blocking buffer = HEPES (pH 7.0, 50 mM), NaCl (150 mM), MnCl (0.1 mM) with 2% (v/v) BSA.

The phage ELISA was performed as described previously (section 00).

**Construction of D1/D2-pep fusions Str1/2/3 and αvβ3-1/2/3.** Primers were designed for each construct using NEBaseChanger™ software to introduce the desired mutation. Site-directed mutagenesis was performed using an NEB Q5® Site-Directed Mutagenesis Kit following the manufacturer's instructions.

Fusion proteins were expressed (section 00), purified using His SpinTrap™ columns (GE Healthcare) and cyclized (section 00) as described previously.

**Octet.** D1/D2 fusion samples were run using a Basic Kinetic Experiment setup on an Octet® RED384 from ForteBio in combination with streptavidin (SA) sensors (ForteBio). Prior to measurement, the biosensor tips were rehydrated for at least 10 min in 200 μL of assay buffer supplemented with 0.1% BSA and 0.02% Tween20 (BuffA). Tilted bottom 384-well plates were used with 50 μL solution per well, which was centrifuged at 1000 RPM for 2 min before measurements were performed. Tips underwent the following steps at 25 °C with 500 Hz: Baseline (BuffA) (60 s), Loading (Antigen at 50 nM) (50 s), Baseline (BuffA) (60 s), Association (D1/D2 fusion proteins ranging from 2 μM to 15.6 nM) (300 s), Dissociation (BuffA) (900 s). Experimental runs were limited to 1 h to minimise evaporation. Raw sensorgrams were

processed in R by first subtracting the background from a reference sample (antigen-loaded, no D1/D2 fusion).

αvβ3 assay buffer: HEPES (pH 7.0, 50 mM), NaCl (150 mM), MnCl (0.1 mM).

**Binding kinetic analysis of synthetic peptides.** The affinity of peptides Str1 and Str1-CCA to streptavidin was measured using the Biacore 8 K (GE Healthcare) at 25 °C.

For cyclization, peptide Str1 (5 mg, GL Biochem) was dissolved in PBS (20 mL) and treated with DTT (1.2 eq.) for 1 h at RT then linker CCA (2 eq.) for 3 h at RT. The reaction was purified by RP-HPLC and characterised by LC-MS to confirm the correct identity of the cyclized product, Str1-CCA. Str1 = 1351 Da and Str1-CCA = 1601 Da.

Streptavidin was covalently immobilised to a CM5 chip surface (Cytiva, BR100530, lot 10340325) using standard amine coupling techniques at a concentration of 16 μg/ml in 10 mM sodium acetate pH 4.5.

The peptides were serially diluted (4.1 – 9000 nM) in HBS-EP + buffer pH 7.4 and flowed over the chip at 30 μL/min, with 180 s association and 600 s dissociation. Multiple buffer-only injections were made under the same conditions to allow for double reference subtraction of the final sensorgram sets, which were analysed using Biacore 8 K Evaluation Software. Each experiment was repeated three times, which gave consistent results. Representative results are presented in the paper.

**Computational modelling.** Modelled structures of peptides Str7-Lin, Str7-DiS and Str7-CCA were constructed using Molecular Operating Environment software (MOE, v 2020.09). Full energy minimisations were run for all peptides using the MMF94x force field available in MOE until the RMSD gradient values were below 0.001. The streptavidin (1DF8) PDB structure was checked for missing atoms and residues using the MOE software before carrying out the docking studies.

The prepared streptavidin structure and lowest energy Str7-CCA peptide conformation were imported into MOE software. Minimising contacts for hydrogen, the structures were subjected to an AMBER94 energy minimisation protocol. The docking energy calculation was carried out within a user-specified three-dimensional docking box (3D docking box) using the simulated annealing method under the MMFF94 force field. The energy grids for docking were generated as grid-based potential fields by the MOE-Dock programme, to reduce the calculation time. Multiple docking poses were generated and analysed to identify the most favourable binding interactions.

**Bioinformatic analysis**

The sequences used for each method are depicted in Tables S1 and S2 for Pep and Str peptides, respectively.

Similarity calculations were performed using Python scripts (python version 3.6.13) that are available at https://github.com/ana-laura476/peptide_similarity. Peptide sequences were converted into SMILES strings using the NovoPro peptide SMILES converter PepSMI (https://www.novoprolabs.com/tools/convert-peptide-to-smiles-string. MAP4 fingerprints were calculated using the script available at https://github.com/reymond-group/map4, and the Jaccard distance was obtained with the get_distance() function from tmap/Minhash library, version 1.0.6. Since this last function gives dissimilarity values (0 for identical, 1 for highly dissimilar), it was transformed by applying *1 – Jaccard distance* giving a similarity value (1 for identical, 0 for highly dissimilar) more easily interpretable and comparable. Using Scikit-learn's TfidfVectorizer() function, we counted the occurrence of each amino acid and then employed the Scikit-learn's cosine_similarity() function to compare text similarities of the written peptides sequences (Scikit-learn version 0.24.2 was used).

**Reporting summary**

Further information on research design is available in the Nature Portfolio Reporting Summary linked to this article.

**Data availability**

Code and data to calculate the peptide similarity can be accessed at https://github.com/ana-laura476/peptide_similarity. Supplementary information is available for this paper. Correspondence, requests for materials and data supporting the findings of the study should be addressed to G.J.L.B., gb453@cam.ac.uk. Source data are provided in this paper.

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

## Acknowledgements

L.B. was funded by a BBSRC industrial CASE studentship (AZ; 10046039). Additional funding was provided by the German Research Foundation (DFG) (493006134 to A.V.V), AstraZeneca (AZ; 10045723 to T.J.) and Fundacão para a Ciência e a Tecnologia (2022.09827.BD; studentship to A.L.D.). We also thank Dr Michael Geeson for providing the CPO-PFP derivative.

## Author contributions

Conceptualised the project: L.B., P.R and G.J.L.B. Supervised the project: G.J.L.B. and M.P. Performed the experiments: L.B., A.V.V., A.L.D, T.R., A.S. and T.J. Analysed the data: L.B., S.O'B. and T.V.M. Computational analysis: A.L.D. and T.R. Wrote the paper: L.B. and G.J.L.B.

## Competing interests

L.B., S.O'B., A.S., T.V.M., P.R., and M.P. are or were formerly employed by AstraZeneca plc. All other authors declare no competing interests.
