## [Peer Review File · Nature Communications]

Proximity-driven Site-specific Cyclization of Phage-displayed PeptidesEditorial note: This manuscript has been previously reviewed at another journal that is not operating a transparent peer review scheme. This document only contains reviewer comments and rebuttal letters for versions considered at *Nature Communications*. Mentions of the other journal have been redacted.

REVIEWER COMMENTS

Editorial note: Reviewer 1 provided confidential comments visible only to the editors, stating that the editors should judge the suitability of this study for this journal.

Reviewer #2 (Remarks to the Author):

While I still like the elegant chemistry for peptide cyclization, the authors could not allay my main concerns that their new approach is inferior to established chemistries and approaches for cyclizing peptides on phage and screening cyclic peptide libraries. The reasons for this are still the requirement of a complex additional step (removal of the prepeptide due to Cys in the first position) and the generation of isomers (which I see mainly as a disadvantage, although I agree that this increases diversity). In addition, as mentioned before, the results with the two easy targets are not convincing and the scope of characterization is not sufficient.

Given the limitations of the new method and the results presented, I do not recommend publication in a high impact journal such as Nature Communication. It would be best if the authors presented a convincing application of the method (e.g. binding to a target for which no binding has been reported or much better binding to a target for which peptide ligands have been reported) to show that the new chemistry and method is at least as good as established methods. The authors state that they do not currently have the ability to do this due to a lab move, and I therefore recommend that the authors are allowed additional time to perform phage display selections on new targets to carefully evaluate the performance of the method.

Reviewer #3 (Remarks to the Author):

I was one of the reviewers when the article was submitted to [journal name redacted], The authors took into consideration most of my previously made comments and provided answers that are satisfactory to me.

Point-by-point response (NCOMMS-23-50988-T)

Reviewer#2 (Remarks to the Author):

While I still like the elegant chemistry for peptide cyclization, the authors could not allay my main concerns that their new approach is inferior to established chemistries and approaches for cyclizing peptides on phage and screening cyclic peptide libraries. The reasons for this are still the requirement of a complex additional step (removal of the prepeptide due to Cys in the first position) and the generation of isomers (which I see mainly as a disadvantage, although I agree that this increases diversity).

ACTION: We thank the reviewer for their comments. This Reviewer has concerns regarding the “complexity of our cyclization” approach. Reviewer #2 states that our phage display libraries require “removal of the pre-peptide due to Cys in the first position”. We would like to clarify that a FLAG tag technology, although validated for phage, was only applied to expression and purification of D1/D2-peptide fusions to ensure homogeneity of the product. When generating phage display libraries, the FLAG tag was omitted to streamline the selection procedure. We experienced no difficulties expressing phage-displayed peptides with cysteine in the first position. To avoid confusion, this has been further clarified in the final manuscript.

In addition, as mentioned before, the results with the two easy targets are not convincing and the scope of characterization is not sufficient.

ACTION: The reviewer suggests that further selections should be performed to show “binding to a target for which no binding has been reported”. However, such an effort would shift the aim of this project to a discovery campaign and detracts from the validation of a novel technology, which is the focus of this work.

Given the limitations of the new method and the results presented, I do not recommend publication in a high impact journal such as Nature Communication. It would be best if the authors presented a convincing application of the method (e.g. binding to a target for which no binding has been reported or much better binding to a target for which peptide ligands have been reported) to show that the new chemistry and method is at least as good as established methods. The authors state that they do not currently have the ability to do this due to a lab move, and I therefore recommend that the authors are allowed additional time to perform phage display selections on new targets to carefully evaluate the performance of the method.

ACTION: Please see response to the previous point. We would like to add that the binding of our hot peptides are solely dependent on the presence of CCA crosslinker as no binding is observed for the disulphide-bonded peptides. This data supports the unique structural features of our cyclised peptides (Str1-CCA) relative to conventional cyclised peptides (Str1). This further supports the uniqueness of our new linker approach to provide access to a new chemical space largely inaccessible to linear and disulphide/ covalent linker-cyclized peptide libraries. This is further supported by our cheminformatic analysis that also shows a large dissimilarity of our hits relative to hits of other libraries.

Reviewer #3 (Remarks to the Author):

I was one of the reviewers when the article was submitted to [journal name redacted], The authors took into consideration most of my previously made comments and provided answers that are satisfactory to me.

ACTION: We thank the reviewer for their comments in the first round of review in [journal name redacted] Which helped us improving significantly our work, and for supporting publication.

REVIEWERS' COMMENTS

Reviewer #3 (Remarks to the Author):

The study introduces a novel cyclopropanone-based proximity-driven chemical linker (CCA) that efficiently cyclizes synthetic peptides and phage-displayed peptides without compromising phage infectivity and viability. This linker enables the construction of highly diverse and stable phage display libraries, allowing the selection of high-affinity cyclic peptide binders. The approach is compatible with reducing environments and does not require purification between reduction and cyclization, thus facilitating the discovery of binders for therapeutically relevant targets.

In the modified manuscript, they have clarified most of the concerns raised by the reviewer 2. For this, the authors included additional experiments which significantly improved the clarity.

To me the revision is sufficient and ready for publication.

(No attachment)